# BATCH-SHAPING FOR LEARNING CONDITIONAL CHANNEL GATED NETWORKS

**Babak Ehteshami Bejnordi, Tijmen Blankevoort & Max Welling**
Qualcomm AI Research[*]
Amsterdam, The Netherlands
`{behtesha,tijmen,mwelling}@qti.qualcomm.com`

## ABSTRACT

We present a method that trains large capacity neural networks with significantly improved accuracy and lower dynamic computational cost. We achieve this by gating the deep-learning architecture on a fine-grained-level. Individual convolutional maps are turned on/off conditionally on features in the network. To achieve this, we introduce a new residual block architecture that gates convolutional channels in a fine-grained manner. We also introduce a generally applicable tool *batch-shaping* that matches the marginal aggregate posteriors of features in a neural network to a pre-specified prior distribution. We use this novel technique to force gates to be more conditional on the data. We present results on CIFAR-10 and ImageNet datasets for image classification, and Cityscapes for semantic segmentation. Our results show that our method can slim down large architectures conditionally, such that the average computational cost on the data is on par with a smaller architecture, but with higher accuracy. In particular, on ImageNet, our ResNet50 and ResNet34 gated networks obtain 74.60% and 72.55% top-1 accuracy compared to the 69.76% accuracy of the baseline ResNet18 model, for similar complexity. We also show that the resulting networks automatically learn to use more features for difficult examples and fewer features for simple examples.

## 1 INTRODUCTION

Almost all deep neural networks have a prior that seems suboptimal: All features are calculated all the time. Both from a generalization and an inference-time perspective, this is superfluous. For example, there is no reason to compute features that help differentiate between several dog breeds, if there is no dog to be seen in the image. The necessity of specific features for classification performance depends on other features. We can make use of this natural prior information, to improve our neural networks. We can also exploit this to spend less computational power on simple and more on complicated examples.

This general idea is commonly encapsulated in the terms *conditional computing* (Bengio, 2013) or gating architectures (Sigaud et al., 2016). It is known that models with increased capacity, for example increased model depth (He et al., 2016) or width (Zagoruyko & Komodakis, 2016), generally help increase model performance when properly regularized. However, as models increase in size, so do their training and inference times. This often limits practical applications of deep learning. By conditionally turning parts of the architecture on or off we can train networks with very large capacity while keeping the computational overhead small. The hypothesis is that when training conditionally gated networks, we can train models with a better accuracy/computation cost trade-off than their fully feed-forward counterparts. In addition, conditional computation with channel gating can potentially increase the interpretability of models. For example, gating can allow us to identify input patterns that trigger the response of a single unit in the network.

Several works in recent literature that successfully learn conditional architectures, for example, ConvNet-AIG (Veit & Belongie, 2018) and dynamic channel pruning (Gao et al., 2018). However, the conditionality is often very coarse as in ConvNet-AIG (Veit & Belongie, 2018), or the amount of actual conditional features learned is very minimal as in Gaternet (Chen et al., 2018). We attempt to solve both. Our contributions are as follows:

---

[*]Qualcomm AI Research is an initiative of Qualcomm Technologies Inc.

- We propose a fine-grained gating architecture that turns individual input and output convolutional maps on or off, leading to features that are individually gated. This allows for a better trade-off between computation cost and accuracy than previous work.

- We propose a generally applicable tool named *batch-shaping* that matches the marginal aggregated posterior of a feature in the network to a specified prior. Depending on the chosen prior, networks can match activation distributions to e.g. the uniform distribution for better quantization, or the Gaussian to enforce behavior similar to batch-normalization (Ioffe & Szegedy, 2015). Specifically, in this paper, we apply batch-shaping to help the network learn conditional features. We show that this helps performance by controlling the gates to be more conditional on the input data at the end of training.

- We show state-of-the-art results compared to other conditional computing architectures such as Convnet-AIG (Veit & Belongie, 2018), SkipNet (Wang et al., 2018a), Dynamic Channel Pruning(Gao et al., 2018), and soft-guided adaptively-dropped neural network (Wang et al., 2018b).

## 2 Background and related work

Literature on gating connections for deep neural networks dates back to Hinton (1981). Gating can be seen as a tri-way connection in a neural network (Droniou et al., 2015), where one output can only be 0 and 1. These connections have originally been used to learn transformations between images with gated Restricted Boltzmann Machines as in Memisevic & Hinton (2007). One of the earliest works to apply this to create sparse network architectures is that of a Mixture of Experts proposed by Jacobs et al. (1991).

Several compression methods exist that statically reduce model complexity. Tensor factorization methods (Jaderberg et al., 2014; Zhang et al., 2016) decompose single layers into two more efficient bottleneck layers. Methods such as channel-pruning (He et al., 2017; Molchanov et al., 2017) remove entire input/output channels from the network. Similarly, full channels can be removed during training as in VIBnets (Dai et al., 2018), Bayesian Compression (Louizos et al., 2017) and L0-regularization (Louizos et al., 2018). These methods reduce the overall model capacity while keeping the accuracy as high as possible. Our method allows for higher model capacity while keeping inference times similar to the papers cited above.

Some networks exploit the complexity of each input example to gain performance. Networks such as Branchynet (Teerapittayanon et al., 2016) and Multi-scale dense net (Huang et al., 2018) have early exiting nodes, where complex examples proceed deeper in the network than simpler ones. Our approach also assigns less computation to simpler examples than more complicated examples but has no early-exiting paths. Both methods of inducing less computation can work in tandem, which we leave for future work.

Several works focus on adaptive spatial attention for faster inference. Figurnov et al. (2017) proposed a residual network based model that dynamically adjusts the number of executed layers for different regions of the image. Difficulty-aware region convolutions were proposed (Li et al., 2017) as an irregular convolution, which allow operating the convolution on specific regions of a feature map. Similarly, tiling-based sparse convolution algorithm was proposed (Ren et al., 2018) which yields a significant speed-up in terms of wall-clock time.

Other works exploit similar conditional sparsity properties of networks as this work. ConvNet-AIG (Veit & Belongie, 2018) and SkipNet (Wang et al., 2018a) turn full residual blocks on or off, conditionally dependent on the input. Dynamic Channel Pruning (Gao et al., 2018) turns individual features on or off similar to our approach, but they choose the top-k features instead, akin to Outrageously large neural networks (Shazeer et al., 2017). This approach loses the benefit of being able to trade-off compute for simple and complex examples. Gaternet (Chen et al., 2018) trains a completely separate network to gate each individual channel of the main network. The overhead of this network is not necessary for learning effective gates, and we show better conditionality of the gates than is achieved by this paper, at almost no overhead.

## 3 Batch-shaping

First, we introduce *batch-shaping*, a general method to match the marginal aggregated posterior distribution over any feature in the neural network to a pre-specified prior distribution. In the next paragraph, we will use this to train gates that fire more conditionally, but we see a large potential value of this tool for many other applications such as training auto-encoders or network quantization.

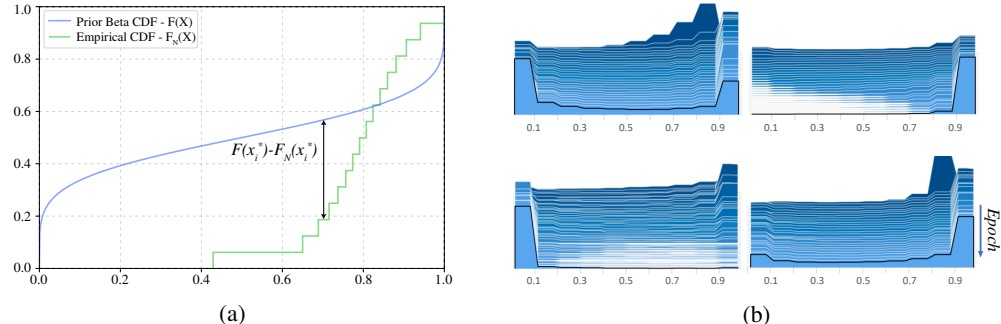

Figure 1: Batch-shaping loss. (a) Illustration of the computation of the batch-shaping loss. (b) The output distribution for four gates and their shapes over different epochs. We can see that initially gates are firing in a conditional pattern, but after removing the shaping loss and introducing the $L_0$-loss they may become fully active, stay conditional or turn off completely.

Consider a parameterized feature in a neural network $X(\theta)$. The intention is to have $X(\theta)$ distributed more according to a chosen probability density function (PDF) $f(x)$, while still being able to differentiate with respect to $\theta$. To do this we consider the Cramér-von-Mises criterion by Anderson (1962), which is a statistical distance between the cumulative distribution function $F(x)$ and an empirical cumulative distribution function $F_N(x)$. The Cramér-von-Mises criterion lends itself naturally to this use-case. Other frequently used statistical distance functions, such as KL-divergence, require calculating a histogram of the samples to compare to $f(x)$, which does not allow for gradients to propagate. As we will see, we can derive gradients with respect to each sample $x$ with the proposed loss function. The Cramér-von-Mises criterion is given by:

$$\omega^2 = \int_{-\infty}^{\infty} \left[ F_N(x) - F(x) \right]^2 \mathrm{d}F(x) \tag{1}$$

We consider batches of $N$ samples $x_{1:N}$ drawn from $X(\theta)$. In order to optimize this we follow Anderson (1962). Sorting $sort(x_{1:N}) = x_{1:N}^*$, replacing $\mathrm{d}F(x)$ with $\mathrm{d}F_N(x)$ and normalizing with $N$ gives us the batch-shaping loss to minimize

$$S(x^*, \lambda) = \frac{\lambda}{N} \sum_{i=1}^{N} \left[ \frac{i}{N+1} - F(x_i^*) \right]^2, \tag{2}$$

where $\lambda$ is a parameter that controls the strength of the loss function. This approach is shown visually in Figure 1a. We can sum this loss for each considered feature in the network to attain a full network batch-shaping loss. Note that we can differentiate $x_{1:N}^*$ with respect to $\theta$ through the sorting operator by keeping the sorted indices. In the backward pass, if a value with index $i$ was sorted to index $j$, we put the error from position $j$ in position $i$. This makes the whole loss term differentiable as long as the chosen CDF function is differentiable. The pseudo-code for the implementation of the Batch-Shaping loss is presented in the Appendix A.

We can use this loss to match the marginal aggregated posterior of a feature in the network to any PDF. For example, if we want to encourage our activations to be Gaussian, we could use the CDF of the Gaussian in the loss function. This could be useful for purposes similar to batch-normalization. Alternatively, the CDF can be that of the uniform distribution, which could help with fixed-point quantization Jacob et al. (2018). We leave this for future work, and only consider a loss function that encourages the conditionality of gates in the next section.

## 4  CHANNEL GATED NETWORKS

In this section we introduce our gating network. While the basic structure of our gating network could be any kind of CNN structure, we use ResNet (He et al., 2016) as the basic structure. Figure 2 shows an overview of a channel gated ResNet block. Formally, a ResNet building block is defined as:

$$x_{l+1} = r(F(x_l) + x_l), \tag{3}$$

Figure 2: Illustration of our channel gated ResNet block and the gating module.

where $x_l \in R^{c^l \times w^l \times h^l}$, and $x_{l+1} \in R^{c^{l+1} \times w^{l+1} \times h^{l+1}}$ denote the input and output of the residual block, and $r$ is the activation function, in this case a ReLU (Nair & Hinton, 2010) function. The residual function $F(x_l)$ is the residual mapping to be learned and is defined as $F = W_2 * r(W_1 * x)$. Here $*$ denotes the convolution operator. $W_1 \in R^{c^l \times c_1^{l+1} \times k \times k}$ is a set of $c_1^{l+1}$ filters, with each filter of size $k \times k$. Similarly, $W_2 \in R^{c_1^{l+1} \times c^{l+1} \times k \times k}$. After each convolution layer, batch normalization (Ioffe & Szegedy, 2015) is used. Our gated residual block is defined as:

$$x_{l+1} = r(W_2 * (G(x_l) \cdot r(W_1 * x_l)) + x_l), \tag{4}$$

where $G$ is a gating module and $G(x_l) = [g_1, g_2, g_3, \cdots, g_{c_1^{l+1}}]$ is the output of the gating function, where $g_c \in \{0, 1\}$: 0 denotes skipping the convolution operation for filter $c$ in $W_1$, and 1 denotes computing the convolution. Here $\cdot$ refers to channel-wise multiplication between the output feature map $r(W_1 * x)$ and the vector $G(x_l)$. The more sparse the output of $G(x_l)$, the more computation we can potentially save.

We chose the position of the gate for two specific reasons. Firstly, the gate is applied *after* the ReLU activation. This prevents the convolution from updating if the gate is off. Placing the gating function before the ReLU caused unstable training behavior. Secondly, we only gate the representation between the two layers in the residual block. We allow each block to use the full input and update the full output. The network only gates each feature that determines how to update the incoming representation. We tested applying multiple gating setups, including before and after each convolutional layer. The proposed setup performed significantly better.

## 4.1 GATING MODULE

To enable a light-weight gating module design, we squeeze global spatial information in $x_l$ into a channel descriptor as input to our gating module, similar to ConvNet-AIG and Squeeze-and-excitation nets (Veit & Belongie, 2018; Hu et al., 2018). This is achieved via channel-wise global average pooling.

For our gating module, we use a simple feed-forward design comprising of two fully connected layers, with only 16 neurons in the hidden layer. We apply batch normalization and ReLU on the output of the first fully connected layer. The second fully connected layer linearly projects the features to unnormalized probabilities $\hat{\pi}_k$, $k \in \{1, 2, \cdots, c_1^{l+1}\}$. We define the logits $\hat{\pi}_k = ln(\pi_k)$.

Our gating module is computationally inexpensive and has an additional overhead that is between $0.018\% - 0.087\%$ of a ResNet block multiply-accumulate (MAC) usage.

To dynamically select a subset of filters relevant for our current input, we need to map the output of our gating module to a binary vector. The task of training binary-valued gates is challenging because we can not directly back-propagate through a non-differentiable gate. In this paper, we leverage a recently proposed approach called Gumbel-Softmax sampling (Jang et al., 2017; Maddison et al., 2017) to circumvent this problem. We consider the binary case of the Gumbel-Max trick, the Binary concrete relaxation $BinConcrete(\pi, \tau)$. In the forward pass we use the discrete argmax and for the backward pass we use a sigmoid function with temperature: $\sigma_\tau(x) = \sigma(\frac{x}{\tau})$. We use $\tau = 2/3$ in all of our experiments as suggested by Maddison et al. (2017).

## 4.2 BATCH-SHAPING BETA DISTRIBUTION PRIOR FOR CONDITIONAL GATES

When initially training the channel-wise gating architecture, many features were trained to be only on or off in the first few epochs. Instead, we would like a feature to be sometimes on and sometimes off

for different data points, to exploit the potential for conditionality. We regulate this by applying the batch-shaping loss, with the CDF of a Beta distribution $I_x(a, b)$, as a prior on each of the gates.

We set $a = 0.6$ and $b = 0.4$ in our experiments, initially inducing 40% sparsity. The Beta distribution will regularize gates towards being sometimes on and sometimes off for different data points, pushing the gates towards the desired batch-wise conditionality. We apply this loss with a strong coefficient $\lambda$ at the start of training to encourage activations to be conditional, and gradually anneal $\lambda$ as training progresses. This allows the network to learn different levels of sparsity or even undo the effect of batch-shaping if necessary. Figure 1b presents the output distribution of four gates during training.

### 4.3 $L_0$-LOSS

Our batch-shaping loss encourages the network to learn more conditional features. However, our actual intention is to find a model that has the best trade-off between (conditional) sparsity and our task loss (e.g., cross-entropy loss). For the second part of our training procedure, we add a loss that regularizes the complexity of the full network explicitly. We use a method proposed by Louizos et al. (2018), which defines a $L_0$ regularization process for neural network sparsification by learning a set of gates with a sparsifying regularizer. Our work can be considered as a conditional version of the $L_0$ gates introduced in this paper, sans stretching parameters. Hence this loss term is a natural choice for sparsifying the activations. We use a modified version of the $L_0$-loss without stretching, defined as:

$$L_C = \gamma \sum_{i=1}^{k} \sigma(ln(\pi_i)), \tag{5}$$

where $k$ is the total number of gates, $\sigma$ is the sigmoid function, and $\gamma$ is a parameter that controls the level of sparsification we want to achieve.

It is essential to mention that introducing this loss too early in training can reduce effective network capacity and potentially hurt performance. If the training procedure is not carefully chosen, the procedure often degenerates into training a smaller architecture, as full convolutional maps are turned off prematurely. Thus, in all our experiments, we introduce this $L_0$-loss after some delay, and we use a warm-up schedule as described in Sønderby et al. (2016)

## 5 EXPERIMENTS

We evaluate the performance of our method on two image classification benchmarks: CIFAR-10 (Krizhevsky, 2009) and ImageNet (Russakovsky et al., 2015). We additionally report preliminary results on the Cityscapes semantic segmentation benchmark (Cordts et al., 2016). For CIFAR-10, we use ResNet20 and ResNet32 architectures (He et al., 2016) as our base model. For ImageNet, we use ResNet18, ResNet34, and ResNet50. We compare our algorithm with competitive conditional computation methods. We additionally perform experiments to understand the learning patterns of the gates and whether they specialize to certain categories. For semantic segmentation, we employ the pyramid scene parsing network (PSPNet) (Zhao et al., 2017) with ResNet-50 backbone.

The training details and hyperparameters for our gated networks trained on CIFAR10, ImageNet, and Cityscapes are provided in the appendix B. The base networks and the gates were trained together from scratch for all of our models.

***CIFAR-10:*** We applied the batch-shaping loss with beta distribution prior from the start of training with a coefficient of $\lambda = 0.75$ and linearly annealed it to zero until epoch 100. Next, the $L_0$-loss was applied to the output of the gates starting from epoch 100, and the coefficient was linearly increased until epoch 300 where it was kept fixed for the rest of the training. For the $L_0$-loss we used $\gamma$ values of $\{0, 1, 2, 5, 10, 15, 20\} \cdot 10^{-2}$ to generate different trade-off points. We also experimented with only using the batch-shaping loss (no $L_0$-loss) with a fixed $\lambda = 0.75$ for the entire training.

We compare our batch-shaped channel gated ResNet20 and ResNet32 models, hereafter referred to as ResNet20-BAS and ResNet32-BAS, with other adaptive computation methods: ConvNet-AIG (Veit & Belongie, 2018), SkipNet (Wang et al., 2018a), and SGAD (Wang et al., 2018b). As shown in Figure 3a, ResNet20-BAS and ResNet32-BAS outperform SkipNet38, SGAD-ResNet32 and ConvNet-AIG variants of ResNet20 and ResNet32 by a large margin. Our results show that given a deep network such as ResNet32, we can reduce the average computation (conditioned on the input) to a value equal or lower than that of a ResNet20 architecture and still achieve better performance. Ideally, a gated ResNet32 model should outperform a gated ResNet20 model at the

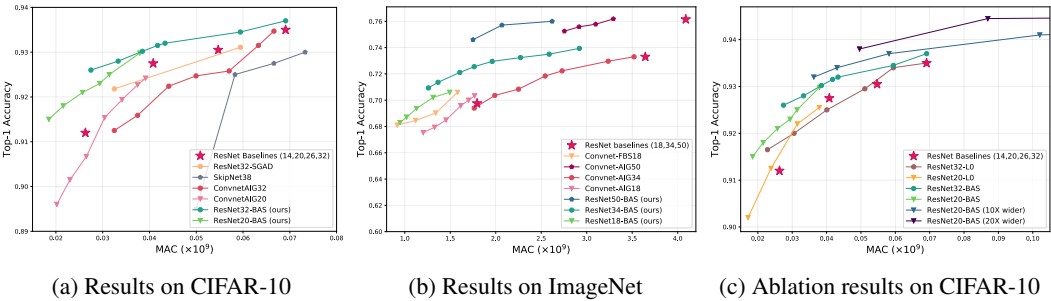

(a) Results on CIFAR-10      (b) Results on ImageNet      (c) Ablation results on CIFAR-10

Figure 3: Comparison of the results of our algorithm and competing methods on (a) CIFAR-10 and (b) ImageNet datasets. (c) shows the effect of the batch-shaping loss on our ResNet20 and ResNet32 gated models trained on CIFAR-10. It also presents the effect of increasing the network's width.

same average computation. This property is evident in our results. However, the competing methods show performance equal to or lower than their smaller sized counterparts, indicating one may instead train a smaller model from scratch.

The highest accuracy obtained by ResNet20-BAS and ResNet32-BAS in Figure 3a relates to the case where we only used the batch-shaping loss with a fixed $\lambda$. Despite the high accuracy, this setting leads to a minor reduction in computation costs. We also experimented with changing the batch-shaping distribution to be more sparse (e.g. changing prior over time). However, the performance degraded significantly. The $L_0$-loss, in contrast, provided a better trade-off between accuracy and MAC saving.

***ImageNet:*** To evaluate the performance of our models on a larger dataset, we applied our gating networks to ImageNet. Similar to CIFAR classification, we introduce the batch-shaping loss from the start of the training with $\lambda = 0.75$ and linearly annealed it to zero until epoch 20. $L_0$-loss was then applied to the output of the gates starting from epoch 30, and the coefficient was linearly increased until epoch 60 where it was kept fixed for the rest of the training. We used $\gamma$ values of $\{0, 1, 2, 5, 10, 15, 20, 30, 40\} \cdot 10^{-2}$ to generate different trade-off points. We also experimented with only using the batch-shaping loss with a fixed $\lambda = 0.75$ for the entire training.

Compared to CIFAR-10, the performance difference with the baseline is larger for ImageNet, likely because of the more substantial complexity of the dataset allowing for more conditionality of the features. We also see an increased performance for lower 'compression rates', similar to what is frequently seen in compression literature because of extra regularization, e.g., as in Frankle & Carbin (2019).

Figure 3b shows the trade-off between the computation cost and Top-1 accuracy for our gated network, ConvNet-AIG, and ConvNet-FBS (Feature Boosting and Suppression) (Gao et al., 2018). The results indicate that our ResNet-BAS models consistently outperform corresponding ConvNet-AIG and ConvNet-FBS models. Similar to the observations on CIFAR-10, the performance of ConvNet-AIG34 degrades to a level lower than that of ConvNet-AIG18, at the same computation cost. Our models, in contrast, make better use of dynamic allocation of features by learning more conditional features. Subsequently, when the average computation cost is on par with ResNet18, our ResNet50-BAS and ResNet34-BAS gated networks achieved 74.40% and 72.55% top-1 accuracy compared to the 70.57% best accuracy of ResNet18-BAS and 69.76% accuracy of the baseline ResNet18 model.

Similar to CIFAR10 experiments, the highest accuracy obtained by batch-shaped models in Figure 3b relates to the case where we only used the batch-shaping loss with a fixed $\lambda$.

***Semantic segmentation:*** For this experiment, we only used our batch-shaping loss with a fixed coefficient. The original PSP network achieves an overall IoU (intersection over union) of 0.706 with a pixel-level accuracy of 0.929 on the validation set. Our gated PSPNet model was able to accomplish an IoU of 0.719 and pixel accuracy of 0.935 while using 76.3% of the MAC count ($\lambda = 0.2$) of the original PSP model. We additionally compared the models when starting training using ImageNet-pretrained weights to initialize the ResNet-50 base network. In this setting, PSPNet achieved an IoU of 0.739 and pixel accuracy of 0.9446. Our gated-model, in comparison, obtained an IoU of 0.744 and pixel accuracy of 0.946 using 76.5% of the PSPNet MAC count ($\lambda = 0.2$). The

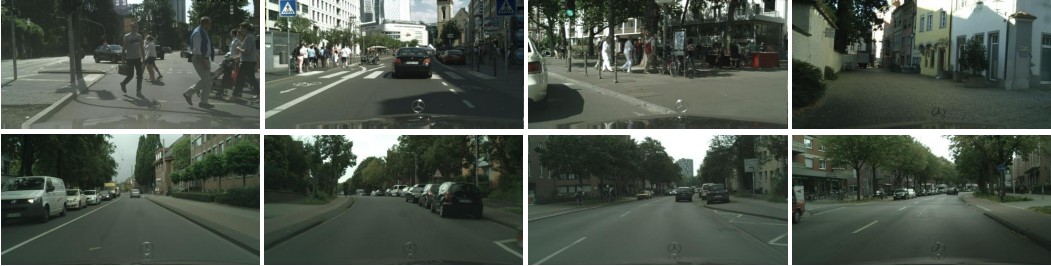

Figure 4: Images with highest MAC usage (top row) and lowest MAC usage (bottom row) from the Cityscapes datasets. The network uses more features for difficult examples and fewer features for simple examples

performance of our gated network further reaches an IoU of 0.747 and pixel accuracy of 0.948 using 95% of the PSPNet MAC count ($\lambda = 0.05$). Figure 4 shows example images from the cityscapes dataset consuming the highest and lowest MAC counts.

## 5.1 EFFECT OF INCREASING WIDTH OF THE NETWORK

To assess the effect of increasing the network's width rather than depth, we trained ResNet20 models on CIFAR10 with per-block width increased with factors of 10X and 20X. The results are shown in Figure 3c. As can be seen, by strongly increasing the number of parameters of the network and allowing for dynamic selection of such parameters, the network consistently achieves higher accuracy.

## 5.2 MODEL INFERENCE-TIME COMPARISON

We compared the inference time of our models to Convnet-AIG (Veit & Belongie, 2018) on CPU in the same setting (see Table 1) for batch-size of one. For batch-computing, a custom convolutional kernel could calculate a mask on the fly. We simulated this for the GPU in the same table. Custom hardware would give us benefits from all MACs saved, including energy savings. See appendix C for further details of our timing setup.

| Model | GPU (ms) | CPU (ms) | Params (total) | MACs (full) | Top-1 Acc |
|---|---|---|---|---|---|
| ResNet18 | **0.46 ± 1.0e-5** | 88.7 ± 8.6e-4 | 11.69M | 1.81G | 0.697 |
| ConvnetAIG34 | 0.71 ± 4.3e-5 | 123.7 ± 0.18 | 19.04M* (21.85M) | 2.73G* (3.66G) | 0.722 |
| ResNet34-BAS | 0.51 ± 5.5e-5 | **86.25 ± 0.22** | **9.15M* (21.91M)** | **1.67G* (3.68G)** | 0.728 |
| ResNet34 | 0.92 ± 3.0e-5 | 149.9 ± 6.5e-4 | 21.79M | 3.66G | 0.733 |
| ConvnetAIG34 (Full) | 0.89 ± 5.8e-5 | 137.75 ± 0.24 | 21.44M* (21.85M) | 3.52G* (3.66G) | 0.732 |
| ResNet34-BAS (Full) | **0.73 ± 1.1e-4** | **111.1 ± 0.36** | **17.77M* (21.91M)** | **2.92G* (3.68G)** | 0.740 |
| ResNet50 | 1.75 ± 3.0e-5 | 184.05 ± 1.8e-4 | 25.55M | 4.09G | 0.761 |
| ConvnetAIG50 | 1.27 ± 4.2e-4 | 142.19 ± 0.09 | 21.97M* (26.56M) | 3.09G* (4.09G) | 0.757 |
| ResNet50-BAS | **1.20 ± 3.4e-4** | **139.82 ± 0.757** | **15.31M* (26.72M)** | **2.07G* (4.11G)** | 0.757 |

Table 1: Inference-time results. Models compared at roughly the same accuracy. Batch-size cpu:1, gpu:128. *Average per example usage on ImageNet val set. CPU uses tensor-slicing, GPU uses custom kernel simulation.

## 5.3 EFFECT OF THE BATCH-SHAPING LOSS

To validate the effectiveness of our proposed batch-shaping loss, we compare the performance of our ResNet-BAS networks in two settings: 1) using both the batch-shaping loss and the $L_0$ complexity loss for training the model similar to the experiments above, or 2) only using the $L_0$ complexity loss.

As can be seen in Figure 3c, the models that additionally use the batch-shaping loss consistently outperform the ones using only the $L_0$ complexity loss. ResNet20-L0 and ResNet32-L0 trained on CIFAR10 appear to have more rapid accuracy degradation than the models trained using the batch-shaping loss. However, our $L_0$-gated models still outperform ConvNet-AIG and SkipNet architectures. This can be attributed to our specific channel gated network architecture which allows for fine-grained dynamic selection of channels or filters in a layer, as compared to the models which are designed to skip whole layers. Very importantly, as shown in Figure 3c, by gating the larger non-BAS ResNet32/38 models, we do not see any improvement over the smaller ResNet20 model

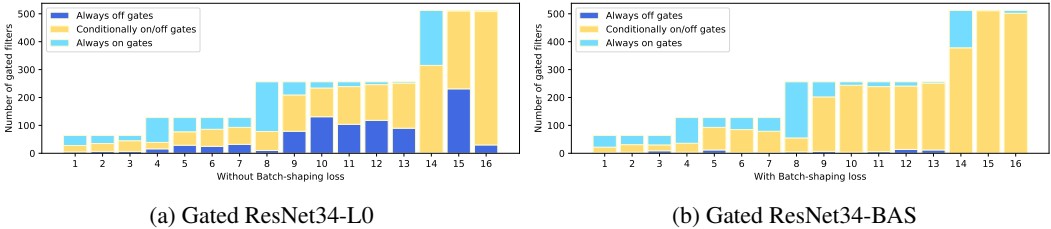

(a) Gated ResNet34-L0                         (b) Gated ResNet34-BAS

Figure 5: The distribution of different gate activation patterns in our ResNet34-L0 and ResNet34-BAS models trained on ImageNet while inducing 60% sparsity. Gates are categorized as always on/off, if they are on/off for more than 99% of the inputs.

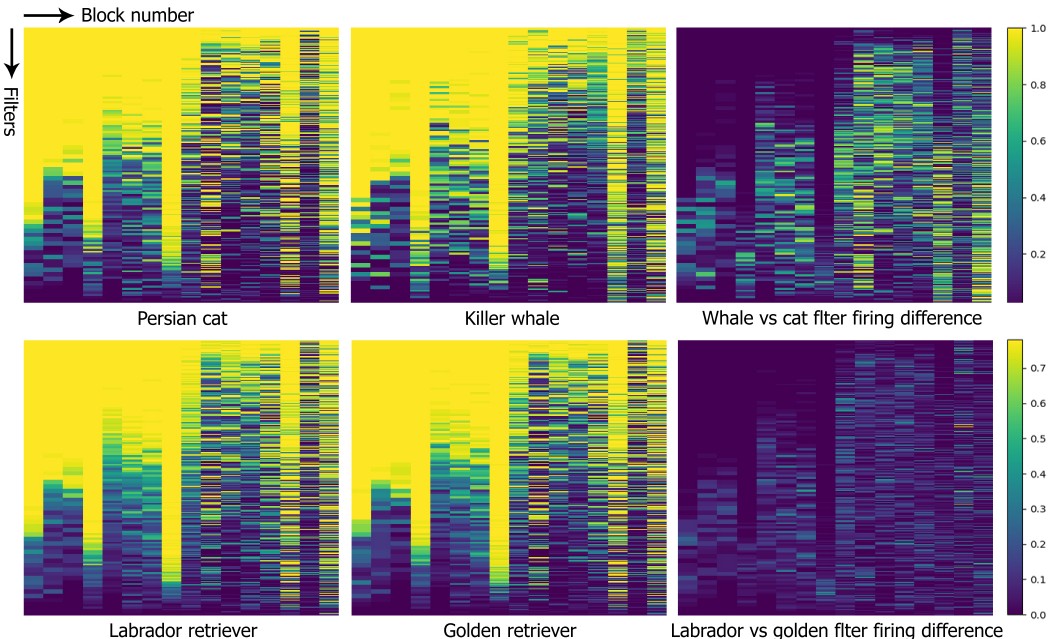

Figure 6: The histogram shows how often individual filters are executed in each layer of a ResNet block (column). For illustration purposes, filters are sorted by execution frequency over the entire validation set. This histogram is computed for our ResNet34-BAS model.

at a similar computation cost. This is in sharp contrast to our ResNet34-BAS model. We observe a similar pattern in ImageNet results (See Figure 8 in appendix D).

## 5.4 GATE DISTRIBUTION

Analyzing the distribution of the learned gates gives us a better insight into the characteristics of the learned features of the network. Ideally, we expect three main gate distributions to appear in a network with dynamic computations: 1) Gates that are always on. We expect certain filters in a network to be of key importance for all types of inputs. 2) Gates that fire conditionally on/off based on the input. The filters that are more input dependent and hence are more specialized for certain categories can be dynamically selected to be executed based on the input. These types of filters are desirable as they contribute to saving computation at the inference phase and formation of conditional expert sub-networks inside our network. Therefore, we would like to maximize the learning of such filters. 3) Gates that are always off. This introduces complete sparsity.

Figure 5 shows the distribution of gates on the ImageNet validation set for our ResNet34-BAS and ResNet34-L0 models. We observe that the majority of the gates activate conditionally for our ResNet34-BAS model. This model prefers conditional sparsity over fully turning gates off. ResNet34-L0 model achieves the same sparsity level by fully turning off a large number of gates.

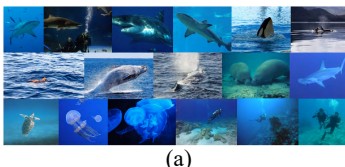 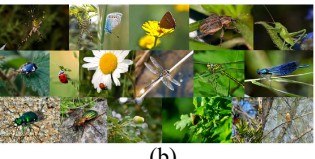 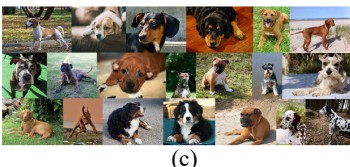

(a)                              (b)                              (c)

Figure 7: Illustration of image categories that activate individual gates in different layers of a ResNet34-BAS. For visualization, we only considered gates that are barely on as they are more specialized and activate for a more specific subset of categories. (a-c) show all possible categories that activate individual gates in the 5th, 10th, and 12th gated ResNet blocks of this model, respectively.

Figure 6 presents an illustration of the fine-grained execution of filters in our gated networks for different image categories. As can be seen, the histograms show small differences in gate execution patterns between similar classes, and large differences between distinct classes.

Figure 7 shows categories that trigger individual gates in different layers of ResNet34-BAS network. Overall, we observe that there are gates at different layers of the network that are activated for a more specific subset of categories. For example, Figure 7a shows that the activation of a gate in the fifth ResNet Block depends on the appearance of large creatures in the water.

## 6 CONCLUSION

In this paper, we presented a fine-grained gating architecture that enables conditional computation in deep networks. Our gating network achieves state-of-the-art accuracy among competing conditional computation architectures on CIFAR10 and ImageNet datasets. In both datasets, given a model with large capacity, our gating method was the only approach that could reduce the inference computation to a value equal or lower than that of a lower capacity base network, while obtaining a higher accuracy. On ImageNet, our ResNet50-BAS and ResNet34-BAS improve the accuracy by more than 4.8% and 2.8% over a ResNet18 model at the same computation cost.

We also proposed a novel batch-shaping loss that can match the marginal aggregated posterior of a feature in the network to any prior PDF. We use it to enforce each gate in the network to be more conditionally activated at the start of training and improve performance significantly. We look forward to seeing many novel applications for this loss in the future, for e.g., autoencoders, better quantized models, and as an alternative to batch-normalization. Another important future research direction that can benefit from our fine-grained gating architecture is continual learning. Designing gating mechanisms that can dynamically decide to allow or prevent the flow of gradients through certain parts of the network could potentially mitigate catastrophic forgetting. Finally, with our gating setup, we could distill smaller sub-networks that work on only a subset of the trained classes.

### ACKNOWLEDGMENTS

The authors would like to thank Markus Nagel, Pim de Haan, and jakub tomczak for their valuable discussions and feedback.

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

## 7 APPENDIX

### A PSEUDO CODE FOR THE IMPLEMENTATION OF THE BATCH-SHAPING LOSS

The pseudo code for the forward and backward pass of the batch-shaping loss. Note that the terms $idx_{sort}$, $\vec{p}_{pdf}$, $\vec{p}_{cdf}$, and $\vec{x}_{sort}$ can be computed and stored in the forward pass and retrieved in the backward pass.

```
Batch-Shaping loss
input  : Gating output, Xm, loss factor γ, Beta
         distribution parameters α, β, batch size
         N
output : Loss for that gating module
loss = 0
for i ← 1 to m do
    x⃗sort = sort(Xi)
    p⃗cdf = BetaCDF(x⃗sort, α, β)
    e⃗cdf = Arrange(1, 1 + N)/(N + 1)
    loss = loss + sum(e⃗cdf − p⃗cdf)²
end
return γ · loss
```

```
Backward pass of the Batch-Shaping loss
input  : Sorted indices idxsort, p⃗pdf, p⃗cdf, and
         e⃗cdf
output : The gradient of the loss with respect to
         the input
for i ← 1 to m do
    gr⃗ad = −2p⃗pdf(e⃗cdf − p⃗cdf)
    gr⃗ad = undo-sort(gr⃗ad)
end
return gr⃗ad
```

### B TRAINING DETAILS

#### B.1 CIFAR10

We trained all models using Nesterov's accelerated gradient descent Nesterov (1983) with a momentum of 0.9 and weight decay factor of $5e^{-4}$. No weight decay was applied on the parameters of the gating modules. We used a standard data-augmentation scheme by randomly cropping and horizontally flipping the images Lin et al. (2013). We trained the models for 500 epochs with a mini-batch of 256. The initial learning rate was 0.1 and it was divided by 10 at epoch 300, 375, and 450.

#### B.2 IMAGENET

We used similar optimization settings to CIFAR-10 with a weight decay factor of $1e^{-4}$. We used a standard data-augmentation scheme adopted from He et al. (2016) and trained the model for 150 epochs with a mini-batch size of 256. All models were trained using a single GPU. The initial learning rate of 0.1 was divided by 10 at epoch 60, 90, and 120.

#### B.3 CITYSCAPES

Cityscapes Cordts et al. (2016) is a dataset for semantic urban scene understanding including 5,000 images with high quality pixel-level annotations and 20,000 additional images with coarse annotations. There are 19 semantic classes for evaluation of semantic segmentation models in this benchmark. For data augmentation, we adopt random mirror and resize with a factor between 0.5 and 2 and use a crop-size of $448 \times 672$ for training. We train and test with only single-scale input and run inference on the whole image. The PSPNet network with the ResNet-50 back-end was trained from scratch with a mini-batch size of 6 for 150k iterations. Momentum and weight decay are set to 0.9 and $1e^{-4}$ respectively. For the learning rate we use similar policy to Chen et al. (2017) where the initial learning rate of $2e - 2$ is multiplied by $(1 - iter_{current}/iter_{max})^{0.9}$. We used the same settings for training our gated PSPNet. Weight decay for the layers in the gating units was set to $1e^{-6}$.

### C DETAILS FOR MODEL TIMING

The output of the gates are computed before the convolutional operations inside a resnet block. The output is fed to the preceding and following convolutional layer kernels, to not compute the masked inputs/outputs. In many practical settings, images are fed to a network one by one (e.g. real-time inference). In this case, the convolutional weights can be loaded from DDR memory to local compute memory conditionally. Computation can be done on sliced tensors, which we implemented in Pytorch.

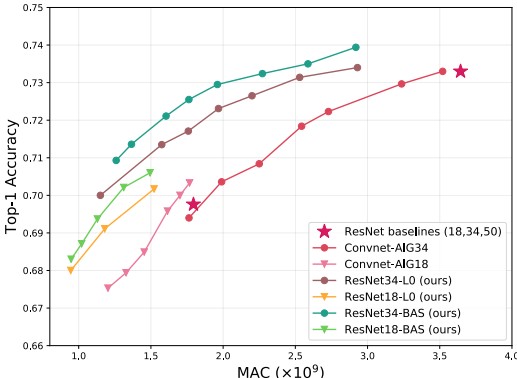

Figure 8: Comparison of the effect of the batch-shaping loss and L0 loss on our ResNet18 and ResNet34 gated models trained on ImageNet

For batch-computing, a custom convolutional kernel could use the calculated mask on the fly. We simulated this for the GPU in the same table. Custom hardware could naturally give us benefits from all MACs saved, including energy savings. All the reported inference times were measured using a machine equipped with an Intel Xeon E5-1620 v4 CPU and an Nvidia GTX 1080 Ti GPU.

### C.1 CPU MEASUREMENTS

Consider $W_1 \in R^{c_1^{in} \times c_1^{out} \times k \times k}$ and $W_2 \in R^{c_2^{in} \times c_2^{out} \times k \times k}$ representing the weight tensors of the first and second layers in a ResNet block, where $c_1^{out} = c_2^{in}$. For each ResNet block, we first use the output of the gates to generate a mask. Using this mask, we slice the original weight tensor of the first layer in the block and apply conv2d on the input featuremap using the sliced weight tensor $W_1 \in R^{c_1^{in} \times c^{slice} \times k \times k}$. We next apply masking for the first batch normalization layer. The input to the second layer is a featuremap with lower number of channels. Using the same mask, we slice the weight tensor of the second layer $W_2 \in R^{c^{slice} \times c_2^{out} \times k \times k}$ and apply the conv2d layer using this tensor.

### C.2 GPU MEASUREMENTS

For the GPU measurements, we first recorded the gating patterns of the entire images in the validation set. For each input image, a sparse model (with a fewer number of convolution kernels in each layer) was defined based on the gating pattern. The computation time was then reported for the sparse model. The overhead caused by the gating modules is included in the wall-time calculation.

### D COMPARISON OF THE EFFECT OF $L_0$ AND BATCH-SHAPING LOSS ON IMAGENET

As can be seen in Figure 8, the models that additionally use the batch-shaping loss consistently outperform the ones using only the $L_0$ complexity loss. However, similar to CIFAR-10 observations, our $L_0$-gated models still outperform ConvNet-AIG.

