# OpenReview forum: "Batch-shaping for learning conditional channel gated networks"
_ICLR.cc/2020/Conference — Accept (Poster)_

### Official Review · AnonReviewer2 · 2019-10-23
**Official Blind Review #2**

**Rating:** 8

**Review:**

The paper describes a method to train a network with large capacity, only parts of which are used at inference time in an input-dependent manner. This leads to accuracy gains without an increase in inference cost. Fine-grained conditional selection is done, using gating of Individual convolutional feature maps. A new method termed “batch shaping” to regularize the network to encourage that the features are used conditionally is introduced and combined with additional regularizer adapted from prior work.

There has been a large body of work along the same research direction. Few of the prior works have focused on fine-grained selection of features, and the ones that have, such as Gao et al, have used a fixed number of features (top-k) across examples instead of dedicating more computation to more difficult examples. In addition, the current work outperforms related prior work through the use of the new regularization technique (batch shaping).

The paper contains thorough comparison to related prior works on three datasets. It also ablates the contribution of the separate aspects of the method -- the fine-grained gating, the batch shaping regularizer, and the L0 penalty. The results demonstrate the all of these aspects contribute to improvements over prior work and result in good accuracy/efficiency trade-offs.

Although this research is not a large departure from prior work, the novelty of the batch shaping regularizer, the thorough empirical study, the experimental gains, and the clarity of the paper makes this a solid contribution.


**Experience Assessment:**

I have read many papers in this area.

**Review Assessment: Checking Correctness Of Derivations And Theory:**

I assessed the sensibility of the derivations and theory.

**Review Assessment: Checking Correctness Of Experiments:**

I assessed the sensibility of the experiments.

**Review Assessment: Thoroughness In Paper Reading:**

I read the paper at least twice and used my best judgement in assessing the paper.

---

> ### Author Response · Authors · 2019-11-12
> **Response to Reviewer #2**
>
> We thank the reviewer for their review and positive assessment of our work.

---

### Official Review · AnonReviewer3 · 2019-10-25
**Official Blind Review #3**

**Rating:** 6

**Review:**

Summary: This paper studies conditional channel gated networks. The network is designed to disable certain channels depending on the inputs. This can be used to save computation. The idea is built on top of “Convolutional Networks with Adaptive Inference Graphs.” The authors propose the technique of batch shaping to encourage the marginal statistics of the gating to be selective to different inputs. With similar inference time, the gated network can achieve better accuracy it can afford adding more layers in the network.

Detailed comments:
- The results show quite significance difference compared to ConvNet-AIG, which demonstrates that batch-shaping is very helpful. Table-1 show 3% increase in accuracy compared ResNet-18 by using similar inference time. It is good that the paper reports wall-clock time measurements.

- It’s good to see some visualizations in the paper, including the image samples and gate locations. I recommend to move Figure 7 to the main paper. A regular neural network can also be used to visualize the sensitivity of patterns of specific neurons. What would be the qualitative differences?

- 1-2x reduction in MAC is not super impressive, especially taking into consideration of the overhead for gathering the active channels for convolution.

- Figure 3a) plot is cut off on the right. The baselines only have a single point in the plot, I guess it is also valid to simply add/remove layers in the baseline models to generate a curve in the plot.

- ResNet-50-L0 is missing in Figure 3b). It would be better if the plots can be grouped better. Currently there are too many lines and it is hard to understand the differences.

- It would be good to see comparisons to some other alternatives to batch shaping. For example, one can penalize so that the average value is around 0.5 by using a L1 loss |E(x) – 0.5|.

- The ImageNet experiment has a very complicated set-up, where L0 loss is applied in the middle of the training. Is this necessary? How important is this step? What would happen if L0 loss is not applied in ImageNet? And what would happen if L0 loss is applied from the beginning? Why is L0 loss not applied in other experiments (e.g. CIFAR or Cityscapes), will L0 loss be beneficial on these benchmarks as well?

- There are a number of related works on adaptive spatial attention for faster inference, which can be included in the related work section.
1) M.  Figurnov,  M.  D.  Collins,  Y.  Zhu,  L.  Zhang,  J.  Huang,D.  P.  Vetrov,  and  R.  Salakhutdinov. Spatially  adaptive computation  time  for  residual  networks. CVPR, 2017.
2) X. Li, Z. Liu, P. Luo, C. C. Loy, and X. Tang.  Not all pixelsare equal:  Difficulty-aware semantic segmentation via deeplayer cascade. CVPR, 2017.
3) M. Ren, A. Pokrovsky, B. Yang, R. Urtasun. SBNet: Sparse Blocks Network for Fast Inference. CVPR, 2018.

Conclusion: The batch shaping technique introduced in this paper has significant improvement on networks that exploit conditional inference. Further understanding of the effect of L0 loss and other alternative loss function is recommended. My overall rating is weak accept.

**Experience Assessment:**

I have published one or two papers in this area.

**Review Assessment: Checking Correctness Of Derivations And Theory:**

N/A

**Review Assessment: Checking Correctness Of Experiments:**

I assessed the sensibility of the experiments.

**Review Assessment: Thoroughness In Paper Reading:**

I read the paper at least twice and used my best judgement in assessing the paper.

---

> ### Author Response · Authors · 2019-11-13
> **Response to Reviewer #3**
>
> We thank the reviewer for their thoughtful review and constructive suggestions. Responses are included inline:
>
> - We moved Figure 7 to the main paper as suggested. A regular neural network does show similar patterns. However, the interpretability of the model behavior could potentially become easier with gated networks. Training neural networks with gates helps to better align the filters with the actual features, and dynamic allocation of filters encourages the relevance to be distributed only to a limited set of units. Analyzing the execution patterns of the gates can potentially make interpretation easier. For example, gates that are mostly active refer to feature extractors that are general and not task/class dependent. Gates which are very selective and rarely execute are more specialized and refer to features that appear only for specific tasks/classes.
>
> - The gathering of active channels doesn't incur much overhead, as you can see in our CPU tables. GPU may not be the only target device, and it is possible to make a non-gathering implementation of this. We would like to refer the reviewer to our first point in our response to reviewer 1. We tend to disagree with that 2X reduction in MAC is not impressive (E.g. in MobileNet V3 paper, the improvements are below a 2x decrease in latency at the same accuracy as MobileNet V2).
>
> - We added ResNet14 and ResNet26 to create more points for the baseline. The models which are 10X and 20X wider have very large MAC usage and including more points will push the rest of the curves to a small portion in the figure. We still increased the margin so the results of the wider networks are more visible. Please see the revised version in the PDF (Fig3c). We are providing the full version in this link: https://ibb.co/xJPCwrV
>
> - We agree with the reviewer and grouped the curves by giving them similar color tones so they are better matched. We also separated the ablation studies in a separate sub-figure to make the original figures less busy. Please find the improved version in the revised version of our paper. Any suggestions to improve the graphs are welcome from the reviewer.
> We expect the results for the ResNet50-L0 models to follow the same trend. We started the experiments but generating multiple points to show the Accuracy-Mac trade-off will unfortunately not be ready by the rebuttal deadline.
>
> - The L1 loss suggested by the reviewer is actually a loss we tried first. While the performance of the models was worse than training with the L0 loss, we found that setting a target rate as in $|E(x) - 0.5|$ causes an undesirable property. The output distribution of a large number of gates becomes unimodal and centered at 0.5. Such gates are not conditional and act as a random dropout gate. By adding the Gumbel noise and taking the argmax, the gate is half the times on and half the times off and by this, the loss objective is easily minimized. Minimizing this loss, therefore, will not result in the formation of conditional gates. We compared the results achieved by this loss, the L0, and batch-shaping loss for the CIFAR-10 experiments and they can be found in this link: https://ibb.co/r6jMBqq
>
> - Great care was put into the set-up, and there are good reasons for each step. We found that introducing the L0 loss too early in training can permanently deactivate gates (always off) and hinder the learning of useful features. It can basically reduce the effective network capacity very early in the training and potentially hurt performance. The L0 loss is, however, required if we want to save more computation and is closer to the actual objective we're interested in, trading off as much compute conditionally for performance. The conditionality itself is a means to an end, and the L0-loss better reflects the actual practical trade-off. For the semantic segmentation, we primarily wanted to explore if channel gated networks are amenable to the semantic segmentation task and did not focus on saving compute. The experiment setup for CIFAR-10 is however similar to ImageNet and L0 was used during training with the same scheme.
>
> - Thank you for pointing us to these references. We added them to the related work section!

---

### Official Review · AnonReviewer1 · 2019-11-05
**Official Blind Review #1**

**Rating:** 6

**Review:**

This paper's focus is on conditional channel-gated networks. Conventional ConvNets process images by computing all the filters, which can be redundant since not all the filters are necessary for a given image. To eliminate this redundancy, this work aims at computing a channel gating on-the-fly, to determine what filters can be turned off. The core contribution of the paper is to propose a "batch-shaping" technique that regularizes the channel gating to follow a beta distribution. Such regularization forces channel gates to either switch on or off. Combined with l_0 regularization, the proposed training technique improves the performance of channel gating: ResNet trained with this technique can achieve higher accuracy with lower theoretical MACs.

Overall, the paper proposes a simple yet effective trick for training gated networks. The paper is well written, and experiments are sufficient in demonstrating the effectiveness of the method.

The main concern for the paper is whether such granular control on the convolution filters can be practically useful. For Conventional ConvNets whose computation is fixed regardless of the input, scheduling the computation on the hardware static and therefore can be easily optimized. When it comes to dynamic networks, especially at such a granular level, it is not clear whether the theoretical complexity reduction can directly translate to actual efficiency (such as latency) improvement. In section 5.2, the author mentions " We simulated this for the GPU in the same table.". Can you elaborate on how you "simulated" the GPU time? How is the simulation done? How well does it predict the actual implementation? Can you implement an efficient kernel for this and show the actual speedup? For the CPU runtime, can you explain in more detail the experimental setting? Can you report the actual latency improvement against theoretical FLOP reduction? For the result in Table 1, why the result of the original ResNet50 is not reported?

**Experience Assessment:**

I have published one or two papers in this area.

**Review Assessment: Checking Correctness Of Derivations And Theory:**

I assessed the sensibility of the derivations and theory.

**Review Assessment: Checking Correctness Of Experiments:**

I assessed the sensibility of the experiments.

**Review Assessment: Thoroughness In Paper Reading:**

I read the paper thoroughly.

---

> ### Author Response · Authors · 2019-11-12
> **Response to Reviewer #1**
>
> We would like to thank Reviewer 1 for their review and constructive comments. Our responses inline:
>
> - There can be two possible ways to implement this. An inefficient implementation for GPU, would do the gather op as in the CPU case (details of CPU implementation below). However, this would lead to a lot of in-GPU memory movement. An efficient implementation could take the boolean mask in the kernel, and only do computations for the output channels that are active. If the convolutional computation is tiled properly over the output dimension, this should result in an output channel sparse convolution with negligible overhead. This can definitely be done, but the actual implementation for GPUs is beyond the scope of this paper. We would also like to mention that there are a lot of other devices/cores that run neural networks, and we feel our results not only hinge on the availability of a GPU kernel. Many networks on mobile devices are still run on the CPU, where our on-device results hold perfectly. Other devices, like Qualcomm's Snapdragon HVX, can easily use the conditionality, since the amount of parallel computation that is done over many cores is lower, and the memory access more sequential. For many such mobile use-cases, the kernel-implementation of our work has almost no overhead.
>
> - For the GPU measurements, we first recorded the gating patterns of the entire images in the validation set. For each input image, a sparse model (with a fewer number of convolution kernels in each layer) was defined based on the gating pattern. The computation time was then reported for the sparse model. The overhead caused by the gating modules is included in the wall-time calculation.
>
> - CPU implementation: As stated in the paper, the results in the table are for an actual CPU implementation and not simulated. In our experimental setting, the actual latency is shown in the table, compared to theoretical FLOP reduction.
>
> Consider $W_{1} \in R^{c_{1}^{in} \times {c_{1}}^{out} \times k \times k}$ and $W_{2} \in R^{c_{2}^{in} \times {c_{2}}^{out} \times k \times k}$ representing the weight tensors of the first and second layers in a ResNet block, where ${c_{1}}^{out} = c_{2}^{in}$. For each ResNet block, we first use the output of the gates to generate a mask. Using this mask, we slice the original weight tensor of the first layer in the block and apply conv2d on the input featuremap using the sliced weight tensor $W_{1} \in R^{c_{1}^{in} \times c^{slice} \times k \times k}$. We next apply masking for the first batch normalization layer. The input to the second layer is a featuremap with lower number of channels. Using the same mask, we slice the weight tensor of the second layer $W_{2} \in R^{c^{slice} \times {c_{2}}^{out} \times k \times k}$ and apply the conv2d layer using this tensor.
>
> - We added the results and timings of ResNet50 to Table 1.
> Model       | GPU (ms)    |   CPU (ms)       | Params | MACs | Top-1 Acc
> ResNet50 | 1.75±3.0e-5 | 184.05±1.8e-4 | 25.55M | 4.09G | 0.761
>
> * We also added details of our actual CPU implementation and the simulation for GPU timing to the appendix.

---

### Public Comment · ~Shanghua_Gao1 · 2019-10-11
**question about ln(πi) in Eq.5**

Thanks for your excellent work.
The πi is the output of the second fc layer according to the statement "The second fully connected layer linearly projects the features to unnormalized probabilities πk". It seems that πi might be negative. And ln() with negative number cause nan.
How can you avoid this problem?

---

> ### Author Response · Authors · 2019-10-14
> **Response to question about Eq.5**
>
> Thank you for your comment and for catching this notation error. The output of the fully connected layer can indeed take negative values and should be denoted as $\hat{\pi_{k}}$ rather than $\pi_k$. We define the logits $\hat{\pi_{k}}=ln({\pi_k})$. We will correct the notation in the revised version.

---

### Decision · Program_Chairs · 2019-12-19

**Decision:**

Accept (Poster)

**Comment:**

The paper describes a method to train a convolutional network with large capacity, where channel-gating (input conditioned) is implemented - thus, only parts of the network are used at inference time. The paper builds over previous work, with the main contribution being a "batch-shaping" technique that regularizes the channel gating to follow a beta distribution, combined with L0 regularization. The paper shows that ResNet trained with this technique can achieve higher accuracy with lower theoretical MACs. Weakness of the paper is that more engineering would be required to convert the theoretical MACs into actual running time - which would further validate the practicality of the approach.